# Intention to use mobile health in antenatal care service: Among primary health care unit workers, Bench-Sheko zone, southwest Ethiopia

Shimeles Wondimu [1]*, Mohamed J. Abawari[2], Yohannes Kebede[2]

1 Department of public health, Mizan-Aman Health Science College, Mizan-Aman, Ethiopia, 2 Department of Health promotion and Health behavior, Faculty of Public Health, Institute of health, Jimma University, Ethiopia

* shimeleswondimu@gmail.com

## Abstract

The use of mobile health (mHealth) technology has the potential to enhance maternal and child health care, particularly in low-income countries. However, evidence regarding its practicality and effectiveness remains limited. In Ethiopia, research on health workers' intention to adopt mHealth technology for maternal health services is notably scarce, emphasizing the need for thoughtful evaluation and further studies to explore its real-world application. This study aims to assess the intention of primary health care unit workers towards using mobile health technology in antenatal care services in Bench-Sheko Zone, Southwest Ethiopia. A cross-sectional study was conducted from June 27, 2023, to July 27, 2023. Data were collected from 316 primary health care unit workers using a simple random sampling technique. SPSS version 25 was used for data analysis, including multivariable linear regression modeling. The response rate was 98.7%. The mean age of participants was 29.2 years, and the mean score for behavioral intention to use mobile technology in ANC services was 65%. Predictors included perceived mobile self-efficacy ($\beta$=0.318, p<0.001), perceived compatibility ($\beta$=0.601, p<0.001), mobile ownership ($\beta$=1.173, p=0.041), eHealth training ($\beta$=0.768, p=0.008), and mobile use experience ($\beta$=0.176, p<0.001). Local health managers should facilitate training to boost health workers' mobile self-efficacy, and policymakers should consider the compatibility of mHealth technology with existing practices.

## Author summary

Mobile health (mHealth) technology is increasingly recognized as a tool to improve maternal and child health services, especially in low-resource settings. However, there is limited evidence on whether health workers are willing to adopt

**Data availability statement:** All datasets used and/or analyzed during the current study are included within the manuscript. The raw data are available from the Mizan-Aman health science college Research Ethics Committee upon reasonable request. Requests can be submitted via email to info@mahsc.edu.et. Due to the lack of an appropriate public repository, data access is restricted. Requests will be evaluated by the Mizan-Aman health science Research Ethics Committee on a case-by-case basis.

**Funding:** The author(s) received no specific funding for this work.

**Competing interests:** The authors have declared that no competing interests exist.

such technology in their daily practice. This study explores the intention of primary health care workers in Bench-Sheko Zone, Southwest Ethiopia, to use mobile technology for antenatal care services. Using a survey of 316 health workers, we identified key factors that influence their willingness to adopt mHealth solutions. Our findings show that mobile self-efficacy, perceived compatibility of technology with existing practices, prior mobile use experience, and eHealth training significantly impact adoption intent. These results highlight the importance of training programs and ensuring that mHealth tools align with the needs of health workers. Policymakers and health managers should focus on enhancing digital literacy and providing supportive infrastructure to maximize the benefits of mHealth technology in maternal health care. By addressing these factors, mobile technology can become a valuable tool in improving health service delivery and outcomes for mothers and newborns in Ethiopia and similar settings.

## Introduction

Mobile health (mHealth) is a component of eHealth, defined by the WHO as medical and public health practice supported by mobile devices [1]. mHealth is used for educating consumers about preventive healthcare, disease surveillance, treatment support, epidemic outbreak tracking, and chronic disease management [2,3]. The WHO's digital health plan for 2020–2025 aims to enhance global health by promoting scalable, long-term mHealth solutions [4]. Ethiopia's National eHealth strategy prioritizes mHealth to improve access, quality, and efficiency in health systems [5].

The theoretical assumptions underlying the use of mobile technology in health services are often rooted in health behavior change theories and technology adoption theories. Key models include the Technology Acceptance Model (TAM), which focuses on perceived usefulness, ease of use, and user attitudes; the Diffusion of Innovation (DOI) model, which considers factors such as innovation characteristics and social systems; and the Unified Theory of Technology Acceptance and Use (UTAUT), which integrates elements from eight different acceptance models to explain the factors influencing individuals' intentions to use technology. These models have been successful in explaining health professionals' intentions to use mobile health technology [6–10].

The current literature on eHealth models reveals key gaps, particularly in low-resource settings. Chattopadhyay (2010) highlights the perceptions of rural healthcare staff and basic ICT needs but lacks focus on scalability and integration into local systems. Similarly, Li et al. (2010) provide an eHealth readiness framework from an EHR perspective but do not address how these models adapt to dynamic healthcare contexts. The high costs of developing and maintaining mHealth technologies challenge low-income countries, straining budgets and hindering adoption. These financial barriers limit accessibility and sustainability in resource-constrained settings [11]. More research is needed on long-term outcomes and the sociocultural and policy factors affecting eHealth sustainability, especially in resource-limited environments like Ethiopia [12,13].

Ethiopia, a low-income country in Sub-Saharan Africa, operates a three-tier healthcare system encompassing primary, secondary, and tertiary levels of care and has made significant strides in improving access to healthcare over recent decades. Despite these efforts, the country faces substantial challenges, including a shortage of healthcare professionals, limited infrastructure in rural areas, and inequities in service delivery. In response to these challenges, digital health solutions, including telemedicine and mobile health (mHealth), have been identified as potential tools to bridge gaps in service accessibility and quality. Telemedicine, in particular, has shown promise in facilitating remote consultations, improving specialist access in underserved areas, and reducing patient travel burdens. However, its implementation in Ethiopia remains in its early stages, hindered by infrastructural limitations, inconsistent internet connectivity, and a lack of standardized protocols. Simultaneously, mobile phone technology has emerged as a complementary tool, offering a more accessible platform for enhancing maternal and child health services, particularly antenatal care (ANC). The Health Sector Transformation Plan (HSTP), a key national policy, prioritizes digital health technologies, but widespread adoption has been slow, with minimal research on the behavioral intentions of healthcare workers to adopt these innovations. This study seeks to address this critical knowledge gap by exploring the determinants influencing healthcare providers' willingness to use mHealth for ANC, leveraging constructs from established technology acceptance and behavioral frameworks to inform policy and practice.

## Materials and methods

### Study design

A cross-sectional quantitative study was conducted from June 27, 2023, to July 27, 2023, in selected primary health care units in Bench-Sheko Zone, Southwest Ethiopia. The cross-sectional approach was selected due to its effectiveness in capturing a snapshot of participants' attitudes and intentions at a specific point in time, making it well-suited for exploring factors influencing mHealth adoption. The design allowed us to gather data from a large, representative sample of health workers within a short time frame and the design is commonly used in technology adoption research and provided a basis for understanding correlations between key variables and behavioral intentions without manipulating variables.

### Sample size and sampling technique

The sample size was calculated using a single population proportion formula, assuming a 50% proportion of health professionals' intention to use mHealth technology in ANC service, with a 95% confidence interval and a 5% margin of error. Including a 10% non-response rate and a design effect of 1.5, the sample size was 320. Simple random sampling was used.

### Data collection tools and procedures

Data were collected using a self-administered questionnaire designed to capture socio-demographic characteristics, mobile phone access, professional characteristics, and behavioral intention to use mHealth technology. The questionnaire was developed based on constructs from behavioral theories and technology acceptance frameworks, specifically the Technology Acceptance Model (TAM), the Unified Theory of Acceptance and Use of Technology (UTAUT), and the Diffusion of Innovations (DOI) framework. Constructs such as perceived ease of use and perceived usefulness (from TAM), facilitating conditions and social influence (from UTAUT), and perceived compatibility and relative advantage (from DOI) were adapted to suit the study context.

To ensure clarity and consistency, key terms were explicitly defined within the questionnaire. For example, "frequently" was defined as occurring three or more times per-day. Additionally, the response structure was tailored to the constructs being measured. Most items employed a 5-point Likert scale, ranging from "Strongly Disagree" to "Strongly Agree," to assess the intensity of participants' attitudes and perceptions. Dichotomous Yes/No questions were used for specific

binary outcomes, such as mobile phone access and certain professional characteristics. This mixed response format allowed for both nuanced data collection and straightforward categorization where appropriate.

To ensure validity and reliability, the questionnaire underwent a pre-test conducted on 5% of the sample population at Chena Health Centre. This process helped refine the items to improve clarity and contextual relevance. Language adjustments were made based on participant feedback to ensure questions were culturally appropriate and easily understood. Cronbach's alpha values (0.864–0.869) demonstrated strong internal consistency across constructs, while the Kaiser-Meyer-Olkin (KMO) measure of sampling adequacy was 0.803, confirming the data's suitability for factor analysis.

To ensure content validity, experts in the field reviewed the survey for clarity, relevance, and comprehensiveness. Their feedback led to refinements in the questionnaire, including modifications to language and the removal of redundant or ambiguous items. Pre-test participants were provided privacy and adequate time to minimize response bias and enhance the quality of responses.

Data quality was ensured through rigorous supervision, real-time monitoring, and validation of data entry by cross-checking random entries. Sampling procedures were carefully monitored to prevent bias, and missing or inconsistent responses were reviewed and addressed before the final data analysis.

### Data analysis procedures

Descriptive analysis was conducted, followed by simple linear regression to select variables for inclusion in the multiple linear regression analysis. Variables with a p-value < 0.25 were considered for inclusion in the model as part of an exploratory approach, allowing for broader consideration of potential predictors. Statistical significance for the final model was set at p-value < 0.05. The reliability of the analysis was ensured through the Variance Inflation Factor (VIF) to check for multicollinearity and residual analysis to validate assumptions like linearity, homoscedasticity, and normality. Consistency was confirmed through repeated tests, and Cronbach's alpha measured internal consistency, supporting the stability of data interpretation.

### Ethical considerations

Ethical approval was obtained from the Mizan-Aman Health Science College Ethical Review Board, and necessary permissions were obtained from the Bench-Sheko Zonal Health Department and district health offices. Informed consent was obtained from all participants. Participants were assured that they could withdraw from the study at any time during the study period. The privacy of subjects was fully respected during data collection and the dissemination of results. All data were securely stored and accessible only to the research team to ensure confidentiality.

### Results

A total of 316 participants completed the questionnaire and were included in the analysis, yielding a response rate of 98.7%. The mean age (±SD) of the study participants was 29.2 ± 5.1 years. Of the participants, 297 (94%) owned a mobile phone, and among them, 241 (76.3%) owned a smartphone. The average mobile usage experience among the participants was 11.9 years, and 136 (43%) of the participants had received training related to eHealth (mHealth).

### Level score of habit of mobile use, perceived mobile self efficacy and perceived compatibility

The mean score for mobile phone use among participants was 63.6% (SD ± 23.6). About 41.5% of participants stated that using a mobile phone at work is habitual. The mean score for perceived mobile self-efficacy was 70.8% (SD ± 20.5), with 48.1% agreeing that they could use a mobile phone with help and 45.1% expressing confidence in using it independently. The mean score for perceived compatibility was 66.7% (SD ± 22.6), with 42.1% stating that mobile phone use fits their work style and 34.8% agreeing that it integrates well into their workflow (Table 1).

**Table 1. Habit of mobile use, Perceived Mobile Self efficacy and Perceived Compatibility, among Health professionals working at PHC in Bench-Sheko zone, southwest Ethiopia in 2023 G.C.**

**Habit of mobile use**

| Items | Likert scale | | | | |
|---|---|---|---|---|---|
| | Strongly disagree | Disagree | Undecided | Agree | Strongly agree |
| I frequently use mobile phone during my daily life (three or more times per day) | 24 (7.6%) | 43 (13.6%) | 29 (9.2%) | 13 (43.4%) | 83 (26.3%) |
| I feel like I must use mobile phone in my daily activities. | 9 (2.8%) | 27 (8.5%) | 76 (24.1%) | 113 (35.8%) | 91 (28.8%) |
| The use of mobile phone in my daily work has become a habit for me. | 6 (1.9%) | 71 (22.5%) | 23 (7.3%) | 131 (41.5%) | 85 (26.9%) |
| I am addicted in using mobile phone in my daily activities | 41 (13.0%) | 93 (29.4%) | 35 (11.1%) | 71 (22.5%) | 76 (24.1%) |
| Using mobile phone have become natural for me | 22 (7.0%) | 61 (19.3%) | 62 (19.6%) | 105 (33.2%) | 66 (20.9%) |
| **Perceived Mobile Self efficacy** | | | | | |
| I can complete my job using mobile phone even if there was no one around to tell me what to do. | 0(0%) | 45 (14.2%) | 40 (12.7%) | 123 (38.9%) | 108 (34.2%) |
| I can use mobile phone in my daily work if someone else help me | 10 (3.2%) | 50 (15.8%) | 51 (16.1%) | 152 (48.1%) | 53 (16.8%) |
| I could complete my job using mobile phone if I had never used a system like it before | 17 (5.4%) | 56 (17.7%) | 23 (7.3%) | 144 (45.6%) | 76 (24.1%) |
| I could complete the job using mobile phone if I had used similar system before this one to do the same job. | 13 (4.1%) | 26 (8.2%) | 18 (5.7%) | 153 (48.4%) | 106 (33.5%) |
| I could use mobile phone in my daily work, if I had seen someone else using it before trying it my self | 5 (1.6%) | 24 (7.6%) | 35 (11.1%) | 153 (48.4%) | 99 (31.3%) |
| **Perceived Compatibility** | | | | | |
| Using mobile phone is compatible with most aspects of my daily work | 11 (3.5%) | 73 (23.1%) | 59 (18.7%) | 94 (29.7%) | 79 (25.0%) |
| Using mobile phone fits well with the way I like to work | 6 (1.9%) | 49 (15.5%) | 52 (16.5%) | 104 (32.9%) | 105 (33.2%) |
| Using mobile phone are proportionate to my work needs | 3 (0.9%) | 83 (26.3%) | 57 (18.0%) | 97 (30.7%) | 76 (24.1%) |
| Using mobile phone fits into my work style | 6 (1.9%) | 58 (18.4%) | 48 (15.2%) | 110 (34.8%) | 94 (29.7%) |
| I think that using mobile phone fits well with the way I like to perform my works | 3 (0.9%) | 44 (13.9%) | 46 (14.6%) | 133 (42.1%) | 90 (28.5%) |

**Level of behavioral intention to use mobile phone technology for ANC service.**

The study found that 65% of health professionals intend to use mobile technology in ANC services. About 40.2% will always try to use it, and 37% plan to use it within the next three months (Table 2).

**Independent predictors of BI to use mobile health technology for ANC service**

Perceived mobile self-efficacy (β =0.332, p<0.001), perceived compatibility (β =0.590,p<0.001), Training status on eHealth (β =0.142, P=0.008), and Mobile phone use experience (β = 0.176 p<0.001) were statistically significant predictor of intention to use mobile health technology for ANC service (Table 3).

## Discussion

The average behavioral intention score to use mobile health technology among healthcare providers in this study is 3.60±0.872. Saeid et al. reported a similar score of 3.50±0.88, likely due to similar sampling techniques and participant demographics [14]. This contrasts with studies in Asia, where scores were higher: 4.28±0.99 in India and 4.13±0.85 in China

**Table 2. Frequency of behavioral intention to use mobile health technology for ANC service, among Health professionals working at PHC in Bench-Sheko zone, southwest Ethiopia in 2023 G.C.**

**Behavioral intention**

| Items | Scales | | | | |
|---|---|---|---|---|---|
| | SD | Disagree | Undecided | Agree | SA |
| I intend to use mobile health technology for ANC service provision in coming three months | 7 (2.2%) | 62 (19.6%) | 75 (23.7%) | 117 (37.0%) | 55 (17.4%) |
| I predict I will use mobile health technology for ANC service provision in the coming three months | 3 (0.9%) | 69 (21.8%) | 78 (24.7%) | 78 (24.7%) | 88 (27.8%) |
| I plan to use mobile health technology for ANC service provision in the coming three months | 0(0%) | 79 (25.0%) | 72 (22.8%) | 107 (33.9%) | 58 (18.4%) |
| I will always try to use mobile health technology for ANC service provision in the coming three months | 4 (1.3%) | 33 (10.4%) | 75 (23.7%) | 127 (40.2%) | 77 (24.4%) |
| I hope I will use mobile health technology for ANC service provision the coming three months | 5 (1.6%) | 56 (17.7%) | 55 (17.4%) | 99 (31.3%) | 101 (32.0%) |

**Table 3. Independent predictors of behavioural intention to use mobile health technology for ANC service among Health professionals working at PHC in Bench-Sheko zone, southwest Ethiopia in 2023 G.C.**

| Independent variables | Unstandardized Coefficients | Standardized Coefficients | Significance | 95% CI for B | |
|---|---|---|---|---|---|
| | B | Beta | P value | Lower | Upper |
| Perceived mobile self-efficacy | 0.332 | 0.313 | <0.001 | 0.242 | 0.421 |
| Perceived compatibility | 0.590 | 0.613 | <0.001 | 0.506 | 0.675 |
| Mobile phone use experience | 0.176 | 0.146 | <0.001 | 0.102 | 0.249 |
| Training status on eHealth | 0.142 | 0.081 | 0.013 | 0.030 | 0.254 |

by Pan Mingha et al [15,16]. In this study, 49.7% of participants scored above the mean for intention to use mobile technology for ANC services, compared to 38.9% in northwest Ethiopia [17]. Perceived mobile self-efficacy and compatibility positively influenced PHC workers' intention to use mobile health technology, aligning with findings by Jeon et al. and Wu et al [18,19]. Additionally, eHealth training increased behavioral intention scores, consistent with studies indicating that technical support and training significantly impact the adoption of mobile health technology among healthcare professionals [20–22].

The study's finding that 65% of health professionals intended to use mHealth for ANC services aligns with results from India (67%) and Kenya (70%), where ease of use and workflow integration were key factors. However, a lower intention (48%) in Nigeria was linked to poor infrastructure and limited training. Our higher rates can be explained by increased mobile phone ownership and eHealth training (43%) and the higher intention observed in this study can also be attributed to Ethiopia's growing digital health focus, supported by the Health Sector Transformation Plan (HSTP), which emphasizes integrating telemedicine and mHealth to bridge healthcare gaps. Despite limited telemedicine infrastructure, the government's efforts to improve mobile network coverage and encourage digital health innovations may have contributed to these findings [23]. These differences underscore the role of local factors like technology access and professional development in mHealth adoption [15,24]. The other significant challenge in implementing mHealth in low-income countries is the high patient-to-staff ratio, which can overwhelm healthcare workers and reduce their intention to adopt new technologies. Additionally, there is a concern that mHealth may be perceived as an imposition, particularly in settings where staff are not tech-savvy. To address this, mHealth should be integrated as a complementary tool rather than a replacement, with adequate training and technical support to ease its adoption [25].

One of the unexpected finding in this study was the modest effect of mobile use experience (β=0.176) on behavioral intention compared to perceived compatibility (β=0.590). While it was expected that extensive mobile use would increase mHealth adoption, the

results suggest that mobile ownership or usage doesn't ensure readiness for professional use. This may be due to the need for specialized training. Similar studies from LMICs, such as South Africa, show that training and task fit are stronger predictors of mHealth adoption than general experience [21]. The other unexpected finding in this study was training on eHealth (β=0.142, p=0.008) had a smaller-than-expected influence on intention, likely due to insufficient content and duration. This suggests a need for more targeted, practical training programs that emphasize hands-on application over theory, as supported by Jeon et al. (2015), where task-specific training improved adoption [26]. Mobile health technologies, while promising, face several limitations in low-resource settings, including issues related to infrastructure, training, and sustainability. Current research indicates that while mHealth can improve healthcare access, its adoption is often hindered by these challenges, which aligns with the findings of this study, highlighting the need for further research to address these barriers and enhance the effectiveness of mHealth interventions in resource-constrained environments.

The study has some limitations. It is prone to social desirability and response biases, particularly acquiescence bias, as data were based on self-reported. Efforts to minimize these biases included reformulating questions and providing clear instructions. Additionally, the study only identified predictors of health workers' intentions rather than cause-and-effect relationships. The focus on a single geographical area, Bench-Sheko zone, along with the two-month data collection period and relatively small sample size, may limit the generalizability of the findings. Future studies with larger sample sizes, extended data collection periods, and broader geographic coverage are needed to enhance the robustness and applicability of the results and additionally, given the absence of actual usage data in this study, we recommend future research to incorporate measures of actual use alongside behavioral intention. This would provide a more complete understanding of the factors influencing the adoption of mHealth technologies, especially in low-income settings.

## Conclusions and recommendations

The intention to use mHealth technology for antenatal care (ANC) among healthcare workers in Bench-Sheko Zone is influenced by mobile self-efficacy, perceived compatibility, mobile use experience, and eHealth training. To enhance mHealth adoption, future research should include longitudinal studies to track changes in attitudes and mixed-methods approaches for a comprehensive understanding of adoption factors and challenges. Comparative and impact evaluation studies will help assess effectiveness and practical implications.

For practical implementation in Ethiopia, improving eHealth training to address self-efficacy and compatibility is essential. Upgrading technological infrastructure and ensuring reliable internet access in rural areas are critical. Aligning mHealth initiatives with existing practices, involving healthcare workers, and establishing feedback mechanisms will support effective integration and increase acceptance of mHealth technologies.

## Acknowledgments

We thank Mizan-Aman Health Science College, the data collectors, study participants, and Bench-Sheko Zone Health Department for their support.

## Author contributions

**Conceptualization:** Shimeles Wondimu, Yohannes Kebede, Mohamed J. Abawari.

**Supervision:** Shimeles Wondimu.

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
