## [Decision Letter · Decision Letter 0]

22 Aug 2024

PDIG-D-24-00289

Intention to use mobile health technology in antenatal care service: among primary health care unit workers, Benchsheko zone, southwest Ethiopia

PLOS Digital Health

Dear Dr. Kebede,

Thank you for submitting your manuscript to PLOS Digital Health. After careful consideration, we feel that it has merit but does not fully meet PLOS Digital Health's publication criteria as it currently stands. Therefore, we invite you to submit a revised version of the manuscript that addresses the points raised during the review process.

Please submit your revised manuscript within 60 days Oct 21 2024 11:59PM. If you will need more time than this to complete your revisions, please reply to this message or contact the journal office at digitalhealth@plos.org. Please include the following items when submitting your revised manuscript:

We look forward to receiving your revised manuscript.

Kind regards,

Haleh Ayatollahi

Section Editor

PLOS Digital Health

Journal Requirements:

1. We note that your Data Availability Statement is currently as follows: All relevant data are within the manuscript

Additional Editor Comments (if provided):

Reviewers' comments:

Reviewer's Responses to Questions

**Comments to the Author**

1. Does this manuscript meet PLOS Digital Health’s publication criteria ? Is the manuscript technically sound, and do the data support the conclusions? The manuscript must describe methodologically and ethically rigorous research with conclusions that are appropriately drawn based on the data presented.

Reviewer #1: Partly

Reviewer #2: Partly

2. Has the statistical analysis been performed appropriately and rigorously?

Reviewer #1: Yes

Reviewer #2: Yes

3. Have the authors made all data underlying the findings in their manuscript fully available (please refer to the Data Availability Statement at the start of the manuscript PDF file)?

Reviewer #1: No

Reviewer #2: No

4. Is the manuscript presented in an intelligible fashion and written in standard English?

PLOS Digital Health does not copyedit accepted manuscripts, so the language in submitted articles must be clear, correct, and unambiguous. Any typographical or grammatical errors should be corrected at revision, so please note any specific errors here.

Reviewer #1: No

Reviewer #2: Yes

5. Review Comments to the Author

Please use the space provided to explain your answers to the questions above. You may also include additional comments for the author, including concerns about dual publication, research ethics, or publication ethics. (Please upload your review as an attachment if it exceeds 20,000 characters)

Reviewer #1: Review Comments to the Author:

1. Publication Criteria:

Partly: The manuscript addresses an important topic and uses appropriate methods. However, it lacks clarity in the theoretical framework, details on questionnaire validation, and ethical considerations. These areas need to be improved to meet the rigorous methodological and ethical standards required by PLOS Digital Health.

2. Statistical Analysis:

Yes: The statistical analysis, including multivariable linear regression modeling, has been performed appropriately and rigorously.

3. Data Availability:

No: The manuscript does not provide detailed information on data availability or any restrictions on access. According to PLOS Data policy, authors must make all data underlying the findings described in their manuscript fully available without restriction. This needs to be addressed.

4. Language and Presentation:

No: The manuscript is generally intelligible and written in standard English but contains typographical and grammatical errors. Consistency in headings and subheadings and minor formatting issues also need to be corrected.

Major Issues:

Theoretical Framework:

Provide a clearer description of the theoretical framework, including specific behavioral theories and technology acceptance models employed.

Generalizability:

Discuss the limitations of focusing on a single geographical area and how this affects the generalizability of the findings.

Questionnaire Validation:

Provide more details on the validation process for the questionnaire used in the study.

Discussion Section:

Strengthen the discussion by comparing the study's findings with those of similar studies in other low- and middle-income countries.

Discuss any unexpected findings and provide potential explanations.

Conclusion and Recommendations:

Provide more actionable and specific recommendations for future research, such as longitudinal studies and mixed-methods approaches.

Discuss practical implementation of findings in the Ethiopian healthcare context.

Data Availability:

Provide information on data availability and any restrictions on access.

Minor Issues:

Abstract:

Include more specific statistical results for clarity.

Ensure consistency in reporting results.

References:

Ensure all references are complete, relevant, and formatted according to the journal's guidelines.

Formatting:

Ensure consistency in headings and subheadings.

Check for minor typographical errors and grammatical issues throughout the manuscript.

Ethical Considerations:

Add a statement on how participants' confidentiality and data security were ensured during the study.

Reviewer #2: Dear Authors, 

The 'intent to use mobile phones in antenatal care' is a known research topic and there are several reported research. Although the paper is written nicely, it, however lacks the following important matters which make it sound and publishable:

1. Literature review, especially in the light of the gap in it. Following papers you may read and cite to understand the eHealth models.

a) Chattopadhyay S. –"A Framework for Studying Perceptions of Rural Healthcare Staff and Basic ICT Support for e-health Use: An Indian Experience". Telemedicine and e-Health (2010); 16(1): pp. 80-88.

b) Li J, Land L.P.W, Ray P. Chattopadhyay S. –"E-Health Readiness Framework from Electronic Health Records Perspective". International Journal of Internet and Enterprise Management: Special Issue in Healthcare (2010); 6(4): pp. 326-348. 

2. Research design

3. Rigorous test of the quality of the data you have collected, and

4. Reliability of MLR-based analysis.

I, therefore, request you to please submit a revised version in the above highlights.

6. PLOS authors have the option to publish the peer review history of their article (what does this mean? ). If published, this will include your full peer review and any attached files.

**Do you want your identity to be public for this peer review?** For information about this choice, including consent withdrawal, please see our Privacy Policy .

Reviewer #1: No

Reviewer #2: Yes: Subhagata Chattopadhyay

---

## [Decision Letter · Decision Letter 1]

4 Dec 2024

PDIG-D-24-00289R1Intention to use mobile health technology in antenatal care service: among primary health care unit workers, Benchsheko zone, southwest EthiopiaPLOS Digital Health Dear Dr. Kebede, Thank you for submitting your manuscript to PLOS Digital Health. After careful consideration, we feel that it has merit but does not fully meet PLOS Digital Health's publication criteria as it currently stands. Therefore, we invite you to submit a revised version of the manuscript that addresses the points raised during the review process. Please submit your revised manuscript within 60 days Feb 02 2025 11:59PM. If you will need more time than this to complete your revisions, please reply to this message or contact the journal office at digitalhealth@plos.org. Please include the following items when submitting your revised manuscript:* A rebuttal letter that responds to each point raised by the editor and reviewer(s). You should upload this letter as a separate file labeled 'Response to Reviewers '. This file does not need to include responses to any formatting updates and technical items listed in the 'Journal Requirements' section below.* A marked-up copy of your manuscript that highlights changes made to the original version. You should upload this as a separate file labeled 'Revised Manuscript with Track Changes '.* An unmarked version of your revised paper without tracked changes. You should upload this as a separate file labeled 'Manuscript '. If you would like to make changes to your financial disclosure, competing interests statement, or data availability statement, please make these updates within the submission form at the time of resubmission. Guidelines for resubmitting your figure files are available below the reviewer comments at the end of this letter. We look forward to receiving your revised manuscript. Kind regards, Haleh AyatollahiSection EditorPLOS Digital Health Haleh AyatollahiSection EditorPLOS Digital Health Leo Anthony CeliEditor-in-ChiefPLOS Digital Healthorcid.org/0000-0001-6712-6626 **Additional Editor Comments (if provided):****Reviewers' Comments:** Reviewer's Responses to Questions

**Comments to the Author**

1. If the authors have adequately addressed your comments raised in a previous round of review and you feel that this manuscript is now acceptable for publication, you may indicate that here to bypass the “Comments to the Author” section, enter your conflict of interest statement in the “Confidential to Editor” section, and submit your "Accept" recommendation.

Reviewer #2: All comments have been addressed

Reviewer #3: (No Response)

Reviewer #4: All comments have been addressed

2. Does this manuscript meet PLOS Digital Health’s publication criteria ? Is the manuscript technically sound, and do the data support the conclusions? The manuscript must describe methodologically and ethically rigorous research with conclusions that are appropriately drawn based on the data presented.

Reviewer #2: Yes

Reviewer #3: Partly

Reviewer #4: Partly

3. Has the statistical analysis been performed appropriately and rigorously?

Reviewer #2: I don't know

Reviewer #3: Yes

Reviewer #4: Yes

4. Have the authors made all data underlying the findings in their manuscript fully available (please refer to the Data Availability Statement at the start of the manuscript PDF file)?

Reviewer #2: (No Response)

Reviewer #3: Yes

Reviewer #4: Yes

5. Is the manuscript presented in an intelligible fashion and written in standard English?

Reviewer #2: Yes

Reviewer #3: No

Reviewer #4: No

6. Review Comments to the Author

Reviewer #2: Dear Authors,

Greetings.

I have read your paper with a great interest. My comments are as follows:

1. The first sentence in the abstract is little over-emphasized, as promising positive effect of mobile technology on MCH is yet to be proven and accepted by the healthcare community. This sentence can be rephrased by removing 'positive'.

Low-income countries struggle with money.

2. Mobile technology is a costly affair to develop and maintain. Hence, low-income country can be better thought of.

3. Yes, a low-income country usually suffers from the ratio disparity between healthcare staff and the patient population. Healthcare service is a manual intervention, especially in the MCH cases. Therefore, there is an apathy to use mobile technology as not everyone is mobile savvy but a highly skilled staff (reflecting at the last item of the "perceived compatibility". A judicious use is recommended, e.g., as a secondary tool and must not be imposed on them. It may jeopardise the entire healthcare system.

4. An "intent vs actual use" graph would be helpful to understand the actual state of the 'promise'.

4. Two-months data is collected. The sample size is not that great, isn't it?

5. Variables with p-value are expected to be <0.05. Hence, why values <0.25 are taken is not clear to me. Is it a typo and refers to 0.025?

6. Finally, I feel authors must write a paragraph in the "discussion" section on the current state of mobile healthcare and related research in view of 'limitations'.

These are some minor corrections I expect to see in the revised version.

Thank you.

Reviewer #3: Dear authors,

I have reviewed your article entitled ""Intention to Use Mobile Health Technology in Antenatal Care Services: Among Primary Health Care Unit Workers in Benchsheko Zone Southwest Ethiopia." The work presents evidence regarding telemedicine adoption and readiness in Ethiopia. I think that it is important work that helps build and support implementation of telemedicine services in LMICs. However, I do have some questions regarding the work that would determine whether the article is fit for publication in this journal.

My main concern is with the questionnaire development and use. I do see that previous comments have addressed some aspects of this, which is great. I would, however, like to know more about how the questionnaire used for the study was developed. There is some information in the introduction about existing questionnaries, such as TAM, DOI, and UTAUT. But it is unclear if these frameworks were used within the questionnaire that was used in the study, whether they were directly used or adapted for the setting, etc. I do see that there are some grammatical errors in the questionnaire, which leads me to assume the questionnaires may have been adapted. If they were adapted, then the authors should clearly state what questions were adapted, how they were adapted, and why (including any support from other published evidence if applicable). I think the validation details of the questionnaire support it's usage, which I see have been added in. I would also suggest adding in the Methods section an overview of each of the parameters and what they indicate regarding telemedicine usage and adoption. Also, I wonder about some of the terminology utilised in the questionnaires, i.e. question 1's 'frequently' - I think this could be interpreted very differently depending on the person, so were these terms defined or quantified at all for the participants? Additionally, how were the answers structured - was it a Likert scale questionnaire, or Yes/No?

Another general concern would be the lack of background information and discussion regarding the health landscape of Ethiopia and what prior work has been done in this area. Could the authors add more information into the introduction and discussion sections regarding research in this area specifically? I see that there is a paper by Saeid et al cited in the discussion, which the authors say they analyse similar participant demographics. However, there is no citation that I can find for Saeid et al in the reference list, and the corresponding number (14) does not have a DOI and is not easily found online. Can you rectify this and provide more context, especially if they have conducted a similar study to yours? In the introduction, the authors should also consider providing more background on the health landscape of Ethiopia. How far has telemedicine implementation come, are there national programs or policies regarding Ethiopian telemedicine, etc. This would provide the average reader some critical information for your article. Also, I notice that the article seems to focus on ANC, but very little is described about ANC. How do these results pertain to or affect ANC in Ethiopia moving forward?

Some other minor comments:

- Introduction, Paragraph 3 Lines 1-3 can you clarify if this is an issue with the models' abilities to account for low resource settings, or the failure of implementation of eHealth in low resource settings?

- Please do another check for grammar errors, especially in the first paragraph of the results section.

- I'm not sure that I see the value of Figures 1 and 2, they seem to repeat the information described in the writing/tables without adding any additional information. Consider whether they are necessary or not

Overall, I think the research topic is valuable and can offer key insights into Ethiopia, LMICs, and ANC, but some of the methods need to be better clarified in order to determine whether the study has been conducted with rigour.

Reviewer #4: Although the reviewer comments are carefully addressed but still there are some concerns in the manuscript:

1) The grammatical errors still needs to be corrected.

2) The table 2 seems insignificant, so better to provide a proper justification specially for the questions included in the table as all the questions seem similar.

7. PLOS authors have the option to publish the peer review history of their article (what does this mean? ). If published, this will include your full peer review and any attached files.

**Do you want your identity to be public for this peer review?** For information about this choice, including consent withdrawal, please see our Privacy Policy .

Reviewer #2: **Yes: ** Subhagata Chattopadhyay

Reviewer #3: No

Reviewer #4: No

---

## [Decision Letter · Decision Letter 2]

5 Mar 2025

Intention to use mobile health technology in antenatal care service: among primary health care unit workers, Benchsheko zone, southwest Ethiopia

PDIG-D-24-00289R2

Dear Kebede,

We are pleased to inform you that your manuscript 'Intention to use mobile health technology in antenatal care service: among primary health care unit workers, Benchsheko zone, southwest Ethiopia' has been provisionally accepted for publication in PLOS Digital Health.

Best regards,

Haleh Ayatollahi

Section Editor

PLOS Digital Health

**Additional Editor Comments (if provided):**

**Reviewer Comments (if any, and for reference):**

Reviewer's Responses to Questions

**Comments to the Author**

1. If the authors have adequately addressed your comments raised in a previous round of review and you feel that this manuscript is now acceptable for publication, you may indicate that here to bypass the “Comments to the Author” section, enter your conflict of interest statement in the “Confidential to Editor” section, and submit your "Accept" recommendation.

Reviewer #4: All comments have been addressed

Reviewer #5: All comments have been addressed

2. Does this manuscript meet PLOS Digital Health’s publication criteria ? Is the manuscript technically sound, and do the data support the conclusions? The manuscript must describe methodologically and ethically rigorous research with conclusions that are appropriately drawn based on the data presented.

Reviewer #4: Yes

Reviewer #5: Yes

3. Has the statistical analysis been performed appropriately and rigorously?

Reviewer #4: Yes

Reviewer #5: Yes

4. Have the authors made all data underlying the findings in their manuscript fully available (please refer to the Data Availability Statement at the start of the manuscript PDF file)?

Reviewer #4: Yes

Reviewer #5: Yes

5. Is the manuscript presented in an intelligible fashion and written in standard English?

Reviewer #4: Yes

Reviewer #5: Yes

6. Review Comments to the Author

Reviewer #4: The submitted manuscript is much more refined now and can be proceeded for publication.

Reviewer #5: 1. It highlights the significance of mHealth adoption in Ethiopia, highlighting the limited research in this area. The inclusion of key statistical results (e.g., β-values and p-values) enhances the study’s credibility.

2. The abstract concisely and effectively summarizes the study’s background, objectives, methods, key findings, and conclusions in a well-structured manner.

3. This study provides concrete steps for improving mHealth adoption, including training, infrastructure development, and policy alignment.

4. The call for longitudinal and mixed-methods studies adds depth to the research and encourages further exploration.

5. The recommendation to improve internet access in rural areas is valid. Given Ethiopia’s technological limitations, adding alternative strategies (e.g., SMS-based interventions, offline mobile applications) would further strengthen the approach.

6. In conclusion, incorporating the role of governmental agencies, NGOs, and other sector partnerships will drive the readers and policy makers successful mHealth adoption, as it requires multi-sectoral collaboration.

7. PLOS authors have the option to publish the peer review history of their article (what does this mean? ). If published, this will include your full peer review and any attached files.

**Do you want your identity to be public for this peer review?** For information about this choice, including consent withdrawal, please see our Privacy Policy .

Reviewer #4: No

Reviewer #5: **Yes: ** Bala Nimmana
